# Quantum Knowledge in Phase Space

**DOI:** 10.3390/e25081227

**Published:** 2023-08-17

**Authors:** Davi Geiger

**Affiliations:** Courant Institute of Mathematical Sciences, New York University, New York, NY 10012, USA; dg1@nyu.edu

**Keywords:** Bayesian statistics, interference, entanglement, entropy, Kullback–Liebler divergence

## Abstract

Quantum physics through the lens of Bayesian statistics considers probability to be a degree of belief and subjective. A Bayesian derivation of the probability density function in phase space is presented. Then, a Kullback–Liebler divergence in phase space is introduced to define interference and entanglement. Comparisons between each of these two quantities and the entropy are made. A brief presentation of entanglement in phase space to the spin degree of freedom and an extension to mixed states completes the work.

## 1. Introduction

This journal issue celebrates Claude Shannon’s 1948 formulation of “lost information” in phone-line signals [1]. It is curious that when von Neumann asked Shannon how he was getting on with their information theory, Shannon replied (according to [2]) “The theory was in excellent shape, except that he needed a good name for “missing information”. “Why don’t you call it entropy”, von Neumann suggested. “In the first place, a mathematical development very much like yours already exists in Boltzmann’s statistical mechanics, and in the second place, no one understands entropy very well, so in any discussion you will be in a position of advantage”.

A quantification of entanglement and interference in quantum phase space for pure states is proposed through Shannon entropy and related concepts (see Appendix B for a brief review). In order to arrive there a phase space probability density must be derived and leads to the next topic that has captivated much of quantum physics discussions since its first days, namely the role of measurement in physics. Measurements are associated with events in statistics, since through measurements full knowledge of a physical variable is acquired. The role of knowledge in quantum physics is then visited.

### 1.1. Bayesian Knowledge in Quantum Physics

In the field of statistics two views offered by the Bayesian [3] and the frequentist [4,5] divide the experts. Bayesian thinking is based on the idea that probabilities are degrees of belief about the events while the frequentist approach is based on the idea that the probability of an event occurring is equal to the long-run frequency with which that event occurs.

### 1.2. Measurements and Knowledge

A measurement in a statistical theory is modeled as an event, since full knowledge of the variable being measured is acquired.

In quantum physics the outcome of a measurement is associated with eigenvalues of a chosen Hermitian operator. For example, if one measures the position or the spin along the *z*-direction of a state, such a position or spin value becomes an event and the state becomes the eigenstate associated with the measured eigenvalue.

In both classical probability and quantum physics, there are scenarios where it is possible to infer full knowledge of a variable without the direct measurement of such a variable. A simple example in classical probability starts with two balls in a bag, one is red and the other is white. By randomly drawing one of them and upon the knowledge of the outcome, one can infer the other ball color immediately without any further measurement. The prior knowledge that there were two different color balls in the bag is combined with the evidence from the observation to give certainty, immediately, on the other ball’s color. This knowledge acquired about the other ball is not causal since if another player is only told the color of the ball that was left in the bag, the player would immediately infer the color of the ball that was drawn. Reasoning with knowledge in this case is invariant with respect to the time of occurrence. In quantum physics entangled states offer an analogous scenario. In teleportation experiments [6,7,8,9] with entangled fermions, the spin *z*-direction of one fermion can immediately be inferred by measuring the spin *z*-direction component of the other fermion. The prior knowledge that both fermions must have opposite spin combined with the observation allows for such inference. However, such knowledge is not causal, it is acquired by combining the prior knowledge with an observation, like in the classical probability example.

The Bayesian view of quantum physics also says that knowledge associated with a state is subjective to the observer. For example, two observers conducting an experiment of quantum teleportation, even if they have the same prior knowledge of the set up, will have different knowledge about the experiment according to when and where a measurement is obtained. According to the special theory of relativity, information or knowledge cannot be transferred instantaneously and so observers of an entangled pair will possess different knowledge of the variables at the time one of them is measured and so their predictions about the outcome of the other variable will differ, e.g., see [10,11]. These experiments with the entangled pair traveling at long distances suggest that quantum physics is best described as a Bayesian theory. It is worth stressing that a theory of knowledge is not necessarily a theory of cause and consequence.

One may wonder, is there a causal explanation for the quantum phenomena? Revisiting the Bohr vs. Einstein debate, e.g., see [12,13] and references, the view put forward here follows the epistemological view of Bohr: quantum theory is today the best model for predictions. The view put forward here also resonates with the ontological concern of Einstein adding the question: “Is there a causal theory that accounts for the quanta phenomena?” The quest here is for a better understanding of quantum theory as a Bayesian theory and the use of Shannon entropy and related concepts to characterize interference and entanglement. The quest for a causal model is left open.

Here, knowledge and information are meant to be the same thing. There is much work in information in quantum physics, e.g., [14,15,16,17] and references, but they use von Neumann mixed states entropy as a starting point which attributes zero information content for all pure states. In contrast, the starting point and focus here is pure states, the core of quantum physics theory. Mixed states do follow from pure states. The emphasis in using the term knowledge or degree of belief for the probabilities instead of information is that it is the language used in Bayesian theory and it stresses that it is subjective. Yet, the Bayesian view provides all the predictions quantum theory can make today and perhaps one can expand it as discussed next.

### 1.3. Phase Space

From a statistical perspective, randomness associated with a quantum state cannot be fully captured by one operator, afterwards an eigenstate of any one operator will seem to contain no randomness, a measurement by this operator will result with certainty in the eigenvalue of this eigenstate.

A unique aspect of quantum physics, expressed by the uncertainty principle, is that eigenvalues of two non-commuting operators cannot be measured simultaneously. Events cannot occur for such pair of variables simultaneously. These are mutually exclusive quantum variables and so, randomness associated with a state cannot be reduced to zero. To characterize the randomness associated with a state, non-commuting operators are then needed. A special operator is *x* with eigenstates |x〉. Together with the unitary evolution operator, they define the space-time properties of states. In quantum mechanics, the uncertainty principle associated with the operator *x* is derived from the non-commuting property [x,p]=iℏ, where *p* is the momentum operator. The variables *x* and $k=\frac{p}{\hbar }$ are Fourier of each other, where *k* is the spatial frequency variable. The randomness of a state $|\Psi_{t}\rangle$ is then captured in phase space, the space formed by the pair of Fourier variables x,k.

In quantum field theory, a relativistic theory, space becomes the domain variable, not an operator, and the spatial frequency *k* becomes the Fourier domain variable [18]. The phase space is then the space of the two domains of the quantum fields, one being the Fourier of the other. In this case, quantum field operators are rooted on the creation and annihilation operators defined in these two domains. A creation operator can then create a particle at some position or with some spatial frequency. The coefficients in front of these operators replace the role of the wave functions establishing the distribution in phase space. Phase space becomes the space where all randomness of a state is captured in quantum field theory.

In probability theory, any statistics of interest is derived from the joint distribution of all variables, so the joint distribution in phase space is the quantity to be derived. A constraint to events in phase space occurs from the fact that $\psi (x,t)=\langle x|\Psi_{t}\rangle$ and $\phi (k,t)=\langle k|\Psi_{t}\rangle$ are the Fourier transform of each other. If one acquires full knowledge of one of the variables at time *t*, expressed by a Dirac delta distribution, then the other phase space variable must be described by a uniform distribution, indicating maximum entropy in this other variable. It is this constraint of our knowledge about the pair of variables *x* and *k* at time *t* that yields the uncertainty principle as clearly formulated by Robertson [19]. The uncertainty principle suggests that events do occur in a volume element in phase space of size ΔxΔk≥1, forming a coarse representation of the phase space. Thus, with respect to the statistical mechanical view of Gibbs [20], quantum mechanics already have a coarse mechanism built in to describe events in phase space.

Early attempts to create a quantum probability distribution in phase space by Wigner [21] and by Husimi [22] ended up with pseudo-distributions that fail Kolmogorov probability axioms and also have consistency difficulties with special relativity. Thus, the need for a pursue of a new approach.

With a new approach to create a phase space probability density, to be developed in Section 2 capturing knowledge about the phase space variables, we further exploit how such knowledge can be used to characterize quantum states entanglement and interference.

### 1.4. Entropy, Interference, and Entanglement

The previous work [23,24] shows how Shannon entropy of a quantum state in phase space captures the loss of knowledge or the loss of information such a state describes. That work explores the hypothesis that knowledge (or information) cannot be gained in a closed quantum system to account for the time arrow. Here, the objective is to quantify interference and entanglement in terms of information loss or gained.

The Kullback–Liebler divergence (reviewed in Appendix B) is employed to define interference as a loss of information if one replaces the state probability density in phase space by a “classical probability density” in phase space. The Kullback–Liebler divergence is also employed to define entanglement as a loss of information if one replaces a state probability density in phase space by a product of state probability density in phase space. Such quantification of interference and entanglement could help our understanding of physical system evolution, for example by restricting which physical phenomena are allowed according to the gain or loss of interference or entanglement.

Position and spin are degrees of freedom (DoFs) required to specify a quantum state. This paper addresses how knowledge in phase space is quantified for position and spin DOFs.

### 1.5. Paper Organization

The remainder of the paper is organized as follows. In Section 2, the Bayesian formulation of probability density in position-momentum phase space is developed. In Section 3, the quantification of interference in phase space is proposed and compared to the entropy in phase space. In Section 4, the quantification of entanglement in phase space is proposed and compared to the entropy in phase space. Section 5 expands the concept of phase space to spin systems and proposes the quantification of entanglement for spin systems. Such quantification can also be expanded to Qbit technology. We also briefly show the approach to mixed states and compare it to von Neumann entropy. Lastly, concluding remarks are provided in Section 6.

## 2. A Bayesian View of Quantum Phase Space

One question immediately arises, what is the meaning of a joint distribution in phase space given that events in phase space cannot occur?

For quantum physics to be a statistical theory, a joint distribution for two variables of non-commuting operators must not require an event and a conditional distribution for the same variables must not require an event to be given. Instead, the joint distribution will describe the degree of belief about the joint variables and the conditional distribution will describe the degree of belief about a variable given the degree of belief about the other variable. An event associated with any one, but not both, of the variables of the non-commuting operators is still possible.

Quantum physics, as a statistical theory, is best described through the states so that classical logical manipulations such as **or** and **and** become additions and products of states analogous to operations in classical probability theory. For example, in the double slit experiment, according to classical logic an electron can pass through slit 1 **or** slit 2, so the quantum state at slit 1 will add with the quantum state at slit 2 to form the final state. To represent a particle A **and** a particle B, one takes the product of the two quantum states (a state in a product of Hilbert Spaces). After the operations with the state occur, quantum probabilities are then associated with the final state $|\Psi_{t}\rangle$ via the probability density matrix $\rho_{t}=|\Psi_{t}\rangle \langle \Psi_{t}|$.

With this view the following proposition and theorem follows

**Lemma** **1**(Conditional Probability Density). *Given the projection of a state to the position basis, $\psi_{t}\left(x\right)=\langle x|\Psi_{t}\rangle$, then the conditional probability density function in the spatial frequency domain is $\rho \left(k|\psi_{t}\left(x\right)\right)={\left|\langle k|\Psi_{t}\rangle \right|}^{2}$. Also, given $\phi_{t}\left(k\right)=\langle k|\Psi_{t}\rangle$, then the conditional probability density function in the position domain is $\rho \left(x|\phi_{t}\left(k\right)\right)={\left|\langle x|\Psi_{t}\rangle \right|}^{2}$.*

**Proof.** Given the wave function $\psi_{t}\left(x\right)=\langle x|\Psi_{t}\rangle$, we can derive via the inverse Fourier transform the conditional wave function in spatial frequency
(1)$$\begin{array}{cc} \phi_{t}\left(k|\psi_{t}\left(x\right)\right) & =\int dx\;\frac{1}{\sqrt{2\pi }}e^{-ikx}\;\psi_{t}\left(x\right) \end{array}$$Clearly, $\phi_{t}\left(k|\psi_{t}\left(x\right)\right)=\int dx\;\langle k|x\rangle \;\langle x|\Psi_{t}\rangle =\langle k|\Psi_{t}\rangle$, where $\langle k|x\rangle =\frac{1}{\sqrt{2\pi }}e^{-ikx}$. Thus, $\rho \left(k|\psi_{t}\left(x\right)\right)=|\phi_{t}\left(k|\psi_{t}\left(x\right)\right){|}^{2}={\left|\langle k|\Psi_{t}\rangle \right|}^{2}$. Similarly, starting from $\phi_{t}\left(k\right)=\langle k|\Psi_{t}\rangle$ and applying the Fourier transform followed by the magnitude square one obtain $\rho \left(x|\phi_{t}\left(k\right)\right)={\left|\langle x|\Psi_{t}\rangle \right|}^{2}$. □

The conditional distribution $\rho (k|\psi_{t}\left(x\right))$, in general, depends on the entire function ψ(x) and not on any one event in *x*. Note that conditional probabilities do not necessarily describe a cause and consequence relation but rather they are knowledge or information relation.

**Theorem** **1**(Joint Distribution in Phase Space). *Given a state $|\Psi_{t}\rangle$, evolving in time according to some Hamiltonian. Then, the joint distribution in phase space is $\rho_{t}\left(x,k\right)=|\psi_{t}{\left(x\right)|}^{2}\;{\left|\phi_{t}\left(k\right)\right|}^{2}$, where $\psi_{t}\left(x\right)=\langle x|\Psi_{t}\rangle$ and $\phi_{t}\left(k\right)=\langle k|\Psi_{t}\rangle$ are the projection in position basis and spatial frequency basis.*

**Proof.** Considering the state projected in the position domain to be a prior known function $\psi_{t}\left(x\right)=\langle x|\Psi_{t}\rangle$ we obtain the conditional projection of the state in the spatial frequency domain from Lemma 1 to be $\phi_{t}\left(k|\psi_{t}\left(x\right)\right)=\langle k|\Psi_{t}\rangle$. We then have the conditional density as $\rho_{t}\left(k|\psi_{t}\left(x\right)\right)={\left|\langle k|\Psi_{t}\rangle \right|}^{2}$.From Bayes’ theorem applied to the density we have $\rho_{t}\left(x,k\right)=\rho_{t}\left(k|\psi_{t}\left(x\right)\right)\rho_{t}^{x}\left(x\right)={\left|\langle k|\Psi_{t}\rangle \right|}^{2}\;{\left|\langle x|\Psi_{t}\rangle \right|}^{2}$.Clearly, we could have started with the state prior $\phi_{t}\left(k\right)=\langle k|\Psi_{t}\rangle$ and obtained via the Fourier transform the (conditional) projection of the state in position $\psi_{t}\left(x|\phi_{t}\left(k\right)\right)=\langle x|\Psi_{t}\rangle$ and then obtained the same joint distribution. □

In quantum statistics, the joint distribution $\rho_{t}\left(x,k\right)=\rho_{t}^{x}\left(x\right)\;\rho_{t}^{k}\left(k\right)$ is not to be interpreted as the product of two independent random variables since the two distributions $\rho_{t}^{x}$ and $\rho_{t}^{k}$ are not independent from each other, and independent events cannot occur simultaneously.

### Entropy in Phase Space

Geiger and Kedem [23] proposed a quantification of knowledge of a quantum state through Shannon entropy associated with the phase space distribution, namely
(2)$$\begin{array}{c} S_{t}=-\int dx\;dk\;\rho_{t}\left(x,k\right)\;\log\rho_{t}\left(x,k\right)\;. \end{array}$$
They showed various desired properties of this entropy, including it to be invariant to canonical transformations, special relativity and to CPT transformations. We adopt this entropy here for the study of interference and entanglement.

## 3. Interference

Given two states $|\Psi^{A}\rangle$ and $|\Psi^{B}\rangle$ and a general superposition of these two states $|\Psi_{A\;\mathrm{or}\;B}\rangle$ as in Equation (Equation 35). The projections of $|\Psi_{A\;\mathrm{or}\;B}\rangle$ written in polar representation are
(3)$$\begin{array}{cc} \psi (x) & =\langle x|\Psi_{A\;\mathrm{or}\;B}\rangle =\frac{e^{i\nu }}{\sqrt{Z_{P}}}\left(\cos\theta_{1}\;|\psi^{A}\left(x\right)|\;e^{i\xi_{A}\left(x\right)}+\sin\theta_{1}\;\left|\psi^{B}\left(x\right)\right|\;e^{i(\xi_{B}\left(x\right)-\varphi_{1})}\right) \end{array}$$
(4)$$\begin{array}{cc} \phi (k) & =\langle k|\Psi_{A\;\mathrm{or}\;B}\rangle =\frac{e^{i\nu }}{\sqrt{Z_{Q}}}\left(\cos\theta_{1}\;|\phi^{A}\left(k\right)|\;e^{i\chi_{A}\left(k\right)}+\sin\theta_{1}\;\left|\phi^{B}\left(k\right)\right|\;e^{i(\chi_{B}\left(k\right)+\varphi_{1})}\right) \end{array}$$
where |.| is the magnitude value, $\xi_{A,B}\left(x\right)$ are the complex phases associated to the wave functions $\psi_{A,B}\left(x\right)$ and similarly $\chi_{A,B}\left(k\right)$ are the complex phases associated to the wave functions $\phi_{A,B}\left(k\right)$.

The probability densities are then
(5)$$\begin{array}{cc} p(x) & ={\left|\psi \left(x\right)\right|}^{2}=\frac{1}{Z_{P}}\left(p^{c}\left(x\right)+|\psi^{A}\left(x\right)|\;|\psi^{B}\left(x\right)|\sin2\theta_{1}\cos\left(\xi_{A}\left(x\right)-\xi_{B}\left(x\right)-\varphi_{1}\right)\right)\;,\\ q(k) & ={\left|\phi \left(k\right)\right|}^{2}=\frac{1}{Z_{Q}}\left(q^{c}\left(k\right)+|\phi^{A}\left(k\right)|\;|\phi^{B}\left(k\right)|\sin2\theta_{1}\cos\left(\chi_{A}\left(k\right)-\chi_{B}\left(k\right)-\varphi_{1}\right)\right) \end{array}$$
where the normalization constants are $Z_{P}=1+\sin2\theta_{1}\;\int dx(|\psi^{A}\left(x\right)|\;|\psi^{B}\left(x\right)|\cos(\xi_{A}\left(x\right)$$-\xi_{B}\left(x\right)-\varphi_{1}))$ and $Z_{Q}=1+\sin2\theta_{1}\;\int dk|\phi^{A}\left(k\right)|\;|\phi^{B}\left(k\right)|\cos\left(\chi_{A}\left(k\right)-\chi_{B}\left(k\right)-\varphi_{1}\right)$, and
(6)$$\begin{array}{cc} p^{c}\left(x\right) & =cos^{2}\theta_{1}\;|\psi^{A}{\left(x\right)|}^{2}+\sin^{2}\theta_{1}\;{\left|\psi^{B}\left(x\right)\right|}^{2}\\ q^{c}\left(k\right) & =\cos^{2}\theta_{1}\;|\phi^{A}{\left(x\right)|}^{2}+\sin^{2}\theta_{1}\;{\left|\phi^{B}\left(x\right)\right|}^{2} \end{array}$$
are probability densities without the interference terms. The upper-index *c* refers to these probability densities also representing classical statistical combination (weighted average) of probability densities associated to the quantum states A and B.

**Definition** **1**(Interference:). *Given two states $|\Psi^{A}\rangle$, $|\Psi^{B}\rangle$ and their linear superposition $|\Psi_{A\;\mathrm{or}\;B}\rangle$ as in Equation *(Equation 35)*. Interference, I, is the amount of information lost when $\rho^{c}\left(x,k\right)=p^{c}\left(x\right)q^{c}\left(k\right)$ is used to approximate ρ(x,k)=p(x)q(k). It is calculated via the Kullback–Liebler divergence between the phase space probability densities ρ(x,k) and $\rho^{c}\left(x,k\right)$, i.e.,*
(7)$$\begin{array}{cc} I(\theta_{1},\varphi_{1},\;|\Psi^{A}\rangle ,|\Psi^{B}\rangle ) & =D_{KL}\left(p\left(x\right)q\left(k\right)||p^{c}\left(x\right)q^{c}\left(k\right)\right)\\ \; & =CrossS\left(p\left(x\right),p^{c}\left(x\right)\right)-S\left(p\left(x\right)\right)+CrossS\left(q\left(k\right),q^{c}\left(k\right)\right)-S\left(q\left(k\right)\right)\\ \; & =\int dx\;{\left|\psi \left(x\right)\right|}^{2}\;\log\left(1+\frac{|\psi^{A}\left(x\right)|\;|\psi^{B}\left(x\right)|\;\sin2\theta_{1}\cos\left(\xi_{A}\left(x\right)-\xi_{B}\left(x\right)-\varphi_{1}\right)}{\cos^{2}\theta_{1}\;|\psi^{A}{\left(x\right)|}^{2}+\sin^{2}\theta_{1}\;{\left|\psi^{B}\left(x\right)\right|}^{2}}\right)\\ \; & \;+\int dk\;{\left|\phi \left(k\right)\right|}^{2}\;\log\left(1+\frac{|\phi^{A}\left(k\right)|\;|\phi^{B}\left(k\right)|\;\sin2\theta_{1}\cos\left(\chi_{A}\left(k\right)-\chi_{B}\left(k\right)-\varphi_{1}\right)}{\cos^{2}\theta_{1}\;|\phi^{A}{\left(k\right)|}^{2}+\sin^{2}\theta_{1}\;{\left|\phi^{B}\left(k\right)\right|}^{2}}\right)\;, \end{array}$$*where CrossS(p,q) is the cross entropy between probability distributions p and q (see Equation *(Equation 41)*) and the phase space entropy Equation *(Equation 2)* for the phase space distributions Equation *(Equation 5)* is given by*
(8)$$\begin{array}{cc} S(|\Psi_{A\;\mathrm{or}\;B}\rangle ) & =-\int \int dx\;dk\;p\left(x\right)q\left(k\right)\;\log\left(p(x)q(k)\right)=-\int dx\;p\left(x\right)\;\log p\left(x\right)-\int dk\;q\left(k\right)\;\log q\left(k\right) \end{array}$$*As one varies the combination of the two states, the larger I is, the larger the interference contribution to the distribution ρ(x,k)=p(x)q(k).*

There is no interference, i.e., I=0, when
the functions’ support in phase space do not overlap, i.e.,
(9)$$\begin{array}{cc} |\psi^{A}\left(x\right)|\;|\psi^{B}\left(x\right)| & =0;\;\forall x\;,\;\mathrm{and}\;|\phi^{A}\left(k\right)|\;|\phi^{B}\left(k\right)|=0;\;\forall k\;, \end{array}$$the complex phases are aligned up to a constant multiple of $\frac{\pi }{2}$, i.e.,
(10)$$\begin{array}{c} \xi_{A}\left(x\right)-\xi_{B}\left(x\right)-\varphi_{1}=n\frac{\pi }{2};\;n\in Z^{+}\;,\;\mathrm{and}\;\chi_{A}\left(k\right)-\chi_{B}\left(k\right)-\varphi_{1}=m\frac{\pi }{2};\;m\in Z^{+}\;, \end{array}$$either $\theta_{1}=0,\frac{\pi }{2}$, since then there is no superposition of states. This will effectively occur when $\psi^{A}\left(x\right)=\psi^{B}\left(x\right)$.
Also, IFF there is no interference, $P\left(x\right)=p^{c}\left(x\right)$ and $Q\left(x\right)=q^{c}\left(x\right)$. Figure 1c,f,i, illustrates scenarios with each state being a coherent state and not overlapping in neither position nor spatial frequency.

Clearly, one can consider the interference just in position representation or just in spatial frequency representation. However, here, the quantification of the interference in phase space distinguish the case (a) when a projection of superposition of two states in position space does not interfere but the same superposition projection in spatial frequency does interfere, from the case (b) a superposition of two states that do not interfere neither in position nor in spatial frequency.

Figure 1 illustrate some scenarios of two coherent states that can be superposed to investigate inference as discussed next. Figure 2 illustrates some scenarios comparing the Kullback–Liebler divergence (KLD) (Equation 7) and the entropy (Equation 8). The entropy captures the notion of *overlap* of the superposition of states. For example, when both states are similar and superposed, the classical addition of probability densities and the quantum superposition do occur. Then, the KLD and the entropy will be small. However, for cases where there is no overlap between the two states, the classical weighted average distribution is a good approximation to the quantum one, the KLD will be small, but the entropy will be large.

Both concepts may be helpful to characterize the knowledge one has about the superposition of states. The KLD captures the distinction between classical probability and quantum probability, while the entropy captures the concept of lack of overlap of two states in quantum phase space. One advantage of the entropy over the KLD is that one does not need to know the components of the superposition of states to evaluate the entropy.

The role of the phase differences is noticeable, in position and in spatial frequency, as per (Equation 10). In addition, for coherent states, used in the simulations, the difference in phase of the state projection in position basis is the differences in the centers of the state projection in the spatial frequency basis. Also vice versa, the phase difference in spatial frequency is the center difference in the position domain. The periodic range for Δμ is reduced for Figure 2c,d. creating the oscillations in the KLD and entropy. Entropy seems to be a good estimation for the interference behavior when the two states overlap either in spatial frequency or in position. However, the more the overlap in both spaces is reduced the more the two quantities differ in behavior.

## 4. Entanglement

Given two states $|\Psi^{A}\rangle$ and $|\Psi^{B}\rangle$ consider the entangled state $|\Psi_{A\;\mathrm{and}\;B}\rangle$ as described in Equation (A3). Extending the language of probability to the quantum states, each state represent a random object. A quantum Bayesian interpretation of this state is that it reflects a joint state of $|\Psi^{A}\rangle$ and $|\Psi^{B}\rangle$, thus the sub-index AandB. Considering the prior to be the state $|\Psi^{A}\rangle$, then the conditional state-operator, which is the product of a state in one Hilbert space times an operator acting in another Hilbert space, is
(11)$$\begin{array}{cc} |\Psi^{\mathrm{cond}-B/A}\rangle =|\Psi^{B}|\Psi^{A}\rangle & =\frac{e^{i\nu_{2}}}{\sqrt{Z_{2}}}(\cos\theta_{2}\;|\Psi^{A}\rangle ⊗|\Psi^{B}\rangle \langle \Psi^{A}|+e^{i\varphi_{2}}\sin\theta_{2}\;|\Psi^{B}\rangle ⊗I)\;, \end{array}$$
where I is the identity operator acting in the other Hilbert space. The conditional state-operator, with an abuse of state notation, reflects the impacts/change to a state $|\Psi^{B}\rangle$ given the knowledge of the state $|\Psi^{A}\rangle$. This conditional state-operator leads to the joint state by acting on the prior state $|\Psi^{A}\rangle$ as follows
(12)$$\begin{array}{c} |\Psi_{A\;\mathrm{and}\;B}\rangle =|\Psi^{\mathrm{cond}-B/A}\rangle |\Psi^{A}\rangle =\frac{e^{i\nu_{2}}}{\sqrt{Z_{2}}}(\cos\theta_{2}\;|\Psi^{A}\rangle |\Psi^{B}\rangle +e^{i\varphi_{2}}\sin\theta_{2}\;|\Psi^{B}\rangle |\Psi^{A}\rangle )\;. \end{array}$$
associated with joint density operator
(13)$$\begin{array}{cc} \rho^{A\;\mathrm{and}\;B} & =|\Psi_{A\;\mathrm{and}\;B}\rangle \langle \Psi_{A\;\mathrm{and}\;B}|\\ \; & =\frac{1}{Z_{2}}\left(\cos\theta_{2}\;|\Psi^{A}\rangle |\Psi^{B}\rangle +e^{i\varphi_{2}}\sin\theta_{2}\;|\Psi^{B}\rangle |\Psi^{A}\rangle \right)\left(\cos\theta_{2}\;\langle \Psi^{B}|\langle \Psi^{A}|+e^{-i\varphi_{2}}\sin\theta_{2}\;\langle \Psi^{A}|\langle \Psi^{B}|\right)\\ \; & =\frac{1}{Z_{2}}\left(\cos^{2}\theta_{2}\;|\Psi^{A}\rangle |\Psi^{B}\rangle \langle \Psi^{B}|\langle \Psi^{A}|+\sin^{2}\theta_{2}\;|\Psi^{B}\rangle |\Psi^{A}\rangle \langle \Psi^{A}|\langle \Psi^{B}|\right.\\ \; & \left.\frac{1}{2}\sin2\theta_{2}\left(e^{-i\varphi_{2}}\;|\Psi^{A}\rangle |\Psi^{B}\rangle \langle \Psi^{A}|\langle \Psi^{B}|+e^{i\varphi_{2}}\;|\Psi^{B}\rangle |\Psi^{A}\rangle \langle \Psi^{B}|\langle \Psi^{A}|\right)\;\right). \end{array}$$

Clearly, we would have obtained the same joint state had we considered the prior to be $|\Psi^{B}\rangle$ and the conditional state-operator to be $|\Psi^{\mathrm{cond}-A/B}\rangle =|\Psi^{A}|\Psi^{B}\rangle =\frac{e^{i\nu_{2}}}{\sqrt{Z_{2}}}(\cos\theta_{2}\;|\Psi^{A}\rangle ⊗I$$+e^{i\varphi_{2}}\sin\theta_{2}\;|\Psi^{B}\rangle ⊗|\Psi^{A}\rangle \langle \Psi^{B}|)$.

The probability density in position space of the joint state is then
(14)$$\begin{array}{cc} p(x,y) & =\langle x|\langle y|\rho^{A\;\mathrm{and}\;B}|y\rangle |x\rangle =\frac{1}{Z_{2,P}}\left(\cos^{2}\theta_{2}\;|\Psi^{A}{\left(x\right)|}^{2}\;|\Psi^{B}{\left(y\right)|}^{2}+\sin^{2}\theta_{2}\;|\Psi^{B}{\left(x\right)|}^{2}\;{\left|\Psi^{A}\left(y\right)\right|}^{2}\right.\\ \; & \;\left.+\;\sin2\theta_{2}\;\cos\left(\xi_{A}\left(x\right)-\xi_{B}\left(x\right)+\xi_{B}\left(y\right)-\xi_{A}\left(y\right)-\varphi_{2}\right)\;|\Psi^{A}\left(x\right)|\;|\Psi^{B}\left(y\right)|\;|\Psi^{A}\left(y\right)|\;|\Psi^{B}\left(x\right)|\;\right)\;. \end{array}$$

Similarly, the probability density $q(k_{x},k_{y})$ in spatial frequency space is
(15)$$\begin{array}{cc} q(k_{x},k_{y}) & =\langle k_{x}|\langle k_{y}|\rho^{A\;\mathrm{and}\;B}|k_{y}\rangle |k_{x}\rangle =\frac{1}{Z_{2,Q}}\left(\cos^{2}\theta_{2}\;|\Phi^{A}\left(k_{x}\right){|}^{2}\;|\Phi^{B}\left(k_{y}\right){|}^{2}+\sin^{2}\theta_{2}\;|\Phi^{B}\left(k_{x}\right){|}^{2}\;{\left|\Phi^{A}\left(k_{y}\right)\right|}^{2}\right.\\ \; & \;\left.+\;\sin2\theta_{2}\;\cos\left(\chi_{A}\left(k_{x}\right)-\chi_{B}\left(k_{x}\right)+\chi_{B}\left(k_{y}\right)-\chi_{A}\left(k_{y}\right)-\varphi_{2}\right)\;|\Phi^{A}\left(k_{x}\right)\left|\;\right|\Phi^{B}\left(k_{y}\right)\left|\;\right|\Phi^{A}\left(k_{y}\right)\left|\;\right|\Phi^{B}\left(k_{x}\right)|\;\right)\;, \end{array}$$
where $\theta_{2}=\frac{\pi }{4}$ and $\varphi_{2}=0,\pi$ describe bosons and fermions, respectively. However, in various empirical works, specially related to quantum computers, one can trap fermions on different locations so that they do not occupy the same state, and still entangle their spin with any general set of parameters. Thus, in theory, one could prepare two fermions to have different spin states and allow them to combine in phase space freely, described by a general set of parameters above. Expanding this formalism to Qbits there are no restrictions on the set of parameters used.

The phase space entropy (Equation 2) for this joint state becomes
(16)$$\begin{array}{cc} S({|\Psi \rangle }_{2}) & =-\int \int \int \int dx\;dk_{x}\;dy\;dk_{y}\;p\left(x,y\right)\;q\left(k_{x},k_{y}\right)\;\log\left(p\left(x,y\right),q\left(k_{x},k_{y}\right)\right)\;. \end{array}$$The product of states, or the disentangled states, are described by the two-state (Equation 12) with $\theta_{2}=0,\frac{\pi }{2}$, i.e.,
(17)$$\begin{array}{cc} \theta_{2} & =0\;\theta_{2}=\frac{\pi }{2}\\ p^{D1}\left(x,y\right) & =|\Psi^{A}{\left(x\right)|}^{2}\;|\Psi^{B}{\left(y\right)|}^{2}\;p^{D2}\left(x,y\right)=|\Psi^{B}{\left(x\right)|}^{2}\;{\left|\Psi^{A}\left(y\right)\right|}^{2}\\ q^{D1}\left(k_{x},k_{y}\right) & =|\Phi^{A}\left(k_{x}\right){|}^{2}\;|\Phi^{B}\left(k_{y}\right){|}^{2}\;\;q^{D2}\left(k_{x},k_{y}\right)=|\Phi^{B}\left(k_{x}\right){|}^{2}\;{\left|\Phi^{A}\left(k_{y}\right)\right|}^{2} \end{array}$$
where the upper index D1 and D2 indicate two different disentangle states.

**Proposition** **1.**
*Given two states $|\Psi^{A}\rangle$, $|\Psi^{B}\rangle$ and the two-state $|\Psi_{A\;\mathrm{and}\;B}\rangle$, shown in *(Equation 12)*. Refer by $p^{D}\left(x,y\right)q^{D}\left(k_{x},k_{y}\right)$ to any of the disentangled states *(Equation 17)*. The Kullback–Liebler divergence $D_{KL}\left(p\left(x,y\right)q\left(k_{x},k_{y}\right)\left|\right|\;p^{D}\left(x,y\right)q^{D}\left(k_{x},k_{y}\right)\right)$ is invariant to any choice of disentangled states for bosons or fermions, where $\theta_{2}=\frac{\pi }{4}$ and $\varphi_{2}=0,\pi$, respectively.*


**Proof.** The proposition follows from performing the decomposition of the logarithm of products into the sum of logarithms and then using the symmetric properties of fermions and bosons. More precisely,
(18)$$\begin{array}{c} D_{KL}^{D}=-S\left(p,q\right)+\int dx\;dy\;dk_{x}\;dk_{y}\;p\left(x,y\right)q\left(k_{x},k_{y}\right)\;\log\left(p^{D}\left(x,y\right)q^{D}\left(k_{x},k_{y}\right)\right) \end{array}$$
where $D_{KL}^{D}$ is short for $D_{KL}^{D}\left(p\left(x,y\right)q\left(k_{x},k_{y}\right)\left|\right|\;p^{D}\left(x,y\right)q^{D}\left(k_{x},k_{y}\right)\right)$. Then, for D1 we get
(19)$$\begin{array}{cc} D_{KL}^{D1} & =-S\left(p,q\right)+\int dx\;dy\;p\left(x,y\right)\log\left(|\Psi^{A}{\left(x\right)|}^{2}\;{\left|\Psi^{B}\left(y\right)\right|}^{2}\right)+\int dk_{x}\;dk_{y}\;q\left(k_{x},k_{y}\right)\;\log\left(|\Phi^{A}\left(k_{x}\right){|}^{2}\;{\left|\Phi^{B}\left(k_{y}\right)\right|}^{2}\right)\\ \; & =-S\left(p,q\right)+\int dx\;p\left(x\right)\log|\Psi^{A}{\left(x\right)|}^{2}\;+\int dy\;p\left(y\right)\log{\left|\Psi^{B}\left(y\right)\right|}^{2}\\ \; & \;+\int dk_{x}\;q\left(k_{x}\right)\;\log|\Phi^{A}\left(k_{x}\right){|}^{2}\;+\int dk_{y}\;q\left(k_{y}\right)\;\log{\left|\Phi^{B}\left(k_{y}\right)\right|}^{2} \end{array}$$
where p(x) = ∫dyp(x,y), p(y) = ∫dxp(x,y), $q(k_{x})$ = $\int dk_{y}\;q\left(k_{x},k_{y}\right)$, $q(k_{y})$ = $\int dk_{x}\;q\left(k_{x},k_{y}\right)$. Note that the integrals yield the same functions p(.) and q(.) due to the symmetric properties for bosons and fermions.Similarly, for D2 we get
(20)$$\begin{array}{cc} D_{KL}^{D2} & =-S\left(p,q\right)+\int dx\;p\left(x\right)\log|\Psi^{B}{\left(x\right)|}^{2}\;+\int dy\;p\left(y\right)\log{\left|\Psi^{A}\left(y\right)\right|}^{2}\\ \; & \;+\int dk_{x}\;q\left(k_{x}\right)\;\log|\Phi^{B}\left(k_{x}\right){|}^{2}\;+\int dk_{y}\;q\left(k_{y}\right)\;\log{\left|\Phi^{A}\left(k_{y}\right)\right|}^{2} \end{array}$$
and clearly every term here has a perfect match in (Equation 19), e.g., $\int dx\;p\left(x\right)\log|\Psi^{B}{\left(x\right)|}^{2}\;=\int dy\;p\left(y\right)\log{\left|\Psi^{B}\left(y\right)\right|}^{2}\;$ and so $D_{KL}^{D2}=D_{KL}^{D1}$. □

**Definition** **2**(Entanglement:). *Given two states $|\Psi^{A}\rangle$, $|\Psi^{B}\rangle$ and the two-state $|\Psi_{A\;\mathrm{and}\;B}\rangle$, shown in *(Equation 12)*, that when projected in phase space yields the probability density distribution $\rho \left(x,y,k_{x},k_{y}\right)=p\left(x,y\right)q\left(k_{x},k_{y}\right)$ given by *(Equation 14)* and *(Equation 15)*. Entanglement, E, is the amount of information lost when the product of states is used to approximate $\rho (x,y,k_{x},k_{y})$. More formally, for bosons or fermions, where $\theta_{2}=\frac{\pi }{4}$ and $\varphi_{2}=0,\pi$ we have*
(21)$$\begin{array}{cc} E(\varphi_{2}=0,\pi ,|\Psi^{A}\rangle ,|\Psi^{B}\rangle ) & =D_{KL}\left(p\left(x,y\right)q\left(k_{x},k_{y}\right)\left|\right|\;p^{D}\left(x,y\right)q^{D}\left(k_{x},k_{y}\right)\right)\\ \; & =CrossS\left(p\left(x,y\right)q\left(k_{x},k_{y}\right),p^{D}\left(x,y\right)q^{D}\left(k_{x},k_{y}\right)\right)-S\left(p\left(x,y\right)q\left(k_{x},k_{y}\right)\right) \end{array}$$*and when the parameters $\theta_{2}$ and $\varphi_{2}$ are free to vary (as in Qbits) then*
(22)$$\begin{array}{cc} E(\theta_{2},\varphi_{2},\;|\Psi^{A}\rangle ,|\Psi^{B}\rangle ) & =\arg\min_{D1,D2}\left[D_{KL}\left(p\left(x,y\right)q\left(k_{x},k_{y}\right)\left|\right|\;p^{D1}\left(x,y\right)q^{D1}\left(k_{x},k_{y}\right)\right)\right.\\ \; & \;\;\left.D_{KL}\left(p\left(x,y\right)q\left(k_{x},k_{y}\right)\left|\right|\;p^{D2}\left(x,y\right)q^{D2}\left(k_{x},k_{y}\right)\right)\right] \end{array}$$

The entanglement vanishes when



$\theta_{2}=0,\frac{\pi }{2}$

$|\Psi^{A}\rangle =|\Psi^{B}\rangle$.

One comparison of interest is between the entanglement (Equation 21) and the entropy (Equation 16). Figure 3 illustrates some scenarios combining two coherent states where these two quantities are evaluated and a comparisons is made. While the definition of entanglement is through the KLD, the entropy captures a similar behavior and it can be evaluated from the state itself, without having to know what the product of the states would be. Entropy behavior seems to be a good estimation for entanglement. One reason is that the basis functions for a product of *N* Hilbert spaces of the total operators $X=x_{1}\times x_{2}\ldots \times x_{N}$ and *K* is given by the product of states $|x_{1}\rangle \ldots |x_{N}\rangle$ and $|k_{1}\rangle \ldots |k_{N}\rangle$. These are the eigenstates of the operators $x_{1}\ldots x_{N}$ and $k_{1}\ldots k_{N}$. Then the product of coherent quantum states, where each one minimizes the phase space entropy, projected in these bases have lower entropy than the entangled states, which are linear superposition of these products.

The study of the more general case of entanglement (Equation 22), which is applicable to Qbits, is left as future work.

## 5. Entanglement for Spin or Qbit Phase Space

The degrees of freedom (DoFs) in quantum physics specify the wave function (projection of the state in position space) and the spin. Thus, when quantifying our knowledge of a quantum system one must also quantify our knowledge of a spin state. Qbits are like spin in formalism, but with less constraints as the Pauli exclusion principle is no longer required (since other aspects of the complete state may already be identified or are already anti-symmetric). What follows also applies to Qbits.

Let us consider two spin states $|\xi_{A}\rangle$ and $|\xi_{B}\rangle$, each formed with $N^{A}$ and $N^{B}$ spin *s* particles, respectively. A spin state formed from these two states may be in a superposition of any of the total spin magnitudes $s\in \;[s_{N},s_{N}-1,\ldots ,\frac{1}{2}\;\mathrm{mod}\;\left(2\;s_{N},2\right)]$, where $s_{N}=Ns$ and $N=N^{A}+N^{B}$, and is written as
(23)$$\begin{array}{c} |\xi \rangle =\frac{e^{i\nu_{2}}}{\sqrt{Z_{2}}}\left(\cos\theta_{2}\;|\xi^{A}\rangle |\xi^{B}\rangle +e^{i\varphi_{2}}\sin\theta_{2}\;|\xi^{B}\rangle |\xi^{A}\rangle \right)\;. \end{array}$$
The operator $S_{z}$ associated with |ξ〉 is given by
(24)$$\begin{array}{c} S_{z}=S_{z}^{A}⊗I_{2N^{B}}+I_{2N^{A}}⊗S_{z}^{B} \end{array}$$
where $I_{N}$ is the identity of dimension *N* and ⊗ is the exterior product.

As discussed earlier, the position and spatial frequency operators for a product of *N* Hilbert spaces is the product of position operators in each Hilbert space. For spin states, the total $S_{z}$ operator (Equation 24) is not just the product of single particle operator. The set of eigenstates of the operators $S_{z}$ and $S^{2}$ include entangled spin states. Let us refer to the eigenstates of $S_{z}$ and $S^{2}$ as $\left\{|\xi_{s,m_{s}}\rangle ;\;s=s_{N},s_{N}-1,\ldots s^{\min}\;,\;\;m_{s}=-s,\ldots ,s\right\}$, where $s^{\min}=\frac{1}{2}\;\mathrm{mod}\;\left(2\;✶\;s_{N},2\right)$. Thus, the state is written in this basis as
(25)$$\begin{array}{c} |\xi \rangle =\sum_{s=s^{\min}}^{s_{N}}\sum_{m=-s}^{s}\alpha_{s,m}|\xi_{s,m}\rangle \;, \end{array}$$
where $\alpha_{s,m}=\langle \xi_{s,m}|\xi \rangle \in C$ and $1=\sum_{s=s^{\min}}^{s_{N}}\sum_{m=-s}^{s}{\left|\alpha_{s,m}\right|}^{2}$.

The phase space for the spin associated with the operators $S_{z}$ and $S^{2}$ is derived from quantizing the sphere, the surface of the ball with a radius of the spin magnitude $\hbar \;s_{N}$, as developed by the Geometric Quantization (GQ) method, e.g., see [25,26,27]. Geiger and Kedem [24] have proposed this approach to evaluate the entropy of a quantum state which is briefly summarized next.

The conjugate basis to $\left\{|\xi_{s,m}\rangle \right\}$ is {|ϕ〉;[0,2π]}, obtained by identifying the angle ϕ, the rotation angle around the *z*-axis of $e^{-iS_{z}\;\phi }$, as the conjugate operator to $S_{z}$. The spin state |ξ〉 in this basis is
(26)|ξ〉=∫02π|ϕ〉〈ϕ|ξ〉(dϕ=∫02πλ(ϕ)|ϕ〉(dϕ,
where
(27)$$\begin{array}{c} \lambda \left(\phi \right)=\langle \phi |\xi \rangle =\sum_{s=s^{\min}}^{s_{N}}\sum_{m=-s}^{s}\alpha_{s,m}\langle \phi |\xi_{s,m}\rangle =\sum_{s=s^{\min}}^{s_{N}}\sum_{m-s}^{s}\alpha_{s,m}\psi_{s,m}\left(\phi \right)\;. \end{array}$$
and
(28)$$\begin{array}{c} \psi_{s,m}\left(\phi \right)=\left\{\begin{array}{ccc} \frac{1}{\sqrt{2\pi }}e^{i(s+m)\phi }, & \;m\geq 0 & \;(\mathrm{northern}\;\mathrm{hemisphere})\;,\\ \frac{1}{\sqrt{2\pi }}e^{i(-s+m)\phi }, & \;m<0 & \;(\mathrm{southern}\;\mathrm{hemisphere})\;. \end{array}\right. \end{array}$$
The two solutions in (Equation 28) are periodic in ϕ and differ by a phase (gauge) transformation of $e^{-i2s\phi }$.

Thus, for a state |ξ〉 with density matrix ρ=|ξ〉〈ξ|, the probabilities of the phase space are the product of the probabilities $\left\{\rho_{s,m}=\langle \xi_{s,m}|\rho |\xi_{s,m}\rangle =|\alpha_{s,m}{|}^{2}\right\}$ with the probability densities $\left\{\rho \left(\phi \right)=\langle \phi |\rho |\phi \rangle =|\lambda \left(\phi \right){|}^{2}\right\}$. Note that given $\left\{\alpha_{s,m}\right\}$ one obtains {λ(ϕ)} via the predefined set of functions (Equation 28), i.e., one can interpret ρ(ϕ) as a conditional probability density $\rho (\phi /\left\{\alpha_{s,m}\right\})$.

Thus, the entropy (Equation 2) of a spin state |ξ〉 in spin phase space is
(29)S=Sz+Sϕ=−∑s=sminsN∑m=−ssρs,mlnρs,m−∫ρ(ϕ)lnρ(ϕ)(dϕ=−∑s=sminsN∑m=−ss|αs,m|2ln|αs,m|2−∫|λ(ϕ)|2ln|λ(ϕ)|2(dϕ.
The first term is the Shannon entropy capturing the randomness of the spin value along the *z*-axis. The second term is differential entropy capturing the randomness of the spin value in the plane perpendicular to the *z*-axis, i.e., the entropy of the polarization angle ϕ. Geiger and Kedem [24] have shown that this entropy reaches its lowest value zero for the eigenstates of the two operators $S_{z}$ and $S^{2}$. Products of states that are described by the superposition of these eigenstates will have higher entropy.

The densities in spin phase space associated with the product state $|\xi^{A}\rangle |\xi^{B}\rangle$ are derived from the projections
(30)$$\begin{array}{c} |\xi^{A}\rangle |\xi^{B}\rangle =\sum_{s=s^{\min}}^{s_{N}}\sum_{m=-s}^{s}\alpha_{s,m}^{A,B}|\xi_{s,m}\rangle \;\Rightarrow \;\alpha_{s,m}^{A,B}=\langle \xi_{s,m}|\xi^{A}\rangle |\xi^{B}\rangle \;\Rightarrow \;\lambda^{A,B}\left(\phi \right)=\sum_{s=s^{\min}}^{s_{N}}\sum_{m-s}^{s}\alpha_{s,m}^{A,B}\psi_{s,m}\left(\phi \right)\;, \end{array}$$

Extending the work of [24] to also define the Kullback–Liebler divergence between a joint state as described by (Equation 12) and the product of the states $|\xi^{A}\rangle |\xi^{B}\rangle$.

**Definition** **3**(Spin Entanglement:). *Given two spin states $|\xi^{A}\rangle ,|\xi^{B}\rangle$ and the joint state |ξ〉, shown in *(Equation 23)*, that when projected in spin phase space yields the probability density distribution $\rho_{s,m}\left(\phi \right)=|\alpha_{s,m}{|}^{2}\;{\left|\lambda \left(\phi \right)\right|}^{2}$. Spin entanglement, sE, is the amount of information lost when the product of states is used to approximate $\rho_{s,m}\left(\phi \right)$. More precisely,*
(31)$$\begin{array}{cc} sE(\theta_{2},\varphi_{2},\;|\xi^{A}\rangle ,|\xi^{B}\rangle ) & =\arg\min_{\left(A,B),(B,A\right)}\left[D_{KL}\left(|\alpha_{s,m}{|}^{2}{\;|\lambda \left(\phi \right)|}^{2}\;|\;|\alpha_{s,m}^{A,B}{|}^{2}\;{\left|\lambda^{A,B}\left(\phi \right)\right|}^{2}\right)\;,\right.\\ \; & \;\;\left.D_{KL}\left(|\alpha_{s,m}{|}^{2}{\;|\lambda \left(\phi \right)|}^{2}\;|\;|\alpha_{s,m}^{B,A}{|}^{2}\;{\left|\lambda^{B,A}\left(\phi \right)\right|}^{2}\right)\right] \end{array}$$

One comparison of interest to be made is between spin entanglement (Equation 31) and spin entropy (Equation 29). In contrast to the position × spatial frequency phase space entanglement, the spin entropy will be minimized and attain value zero for some entangled states that are eigenstates of the total spin operators, such as it is the case of Bell entangled states for two particles [24]. The product of states will then be described by the superposition of these entangled states and thus, will not have a well-defined spin magnitude. Their entropy will be larger. The KLD will then anti-correlate with the entropy. A more detailed study is left for the future.

### Expansion to Mixed States

One extension of this approach to mixed states starts from the density matrix derived from the general state (Equation 23), namely
(32)$$\begin{array}{cc} \rho =|\xi \rangle \langle \xi | & =\frac{1}{Z_{2}}\left(\cos^{2}\theta_{2}\;|\xi^{A}\rangle |\xi^{B}\rangle \langle \xi^{B}|\langle \xi^{A}|+\sin^{2}\theta_{2}\;|\xi^{B}\rangle |\xi^{A}\rangle \langle \xi^{A}|\langle \xi^{B}|\right.\\ \; & \;\left.+\frac{1}{2}e^{-i\varphi_{2}}\sin2\theta_{2}\;|\xi^{A}\rangle |\xi^{B}\rangle \langle \xi^{A}|\langle \xi^{B}|+\frac{1}{2}e^{i\varphi_{2}}\sin2\theta_{2}\;|\xi^{B}\rangle |\xi^{A}\rangle \langle \xi^{B}|\langle \xi^{A}|\right)\;. \end{array}$$Then, tracing out the density matrix (and assuming the states $|\xi^{A}\rangle ,|\xi^{B}\rangle$ to be in orthogonal Hilbert spaces to each other)
(33)$$\begin{array}{c} \rho_{s}^{\mathrm{Mixed}}=\langle \xi^{A}||\xi_{s}\rangle \langle \xi_{s}||\xi^{A}\rangle +\langle \xi^{B}||\xi_{s}\rangle \langle \xi_{s}||\xi^{B}\rangle =\cos^{2}\theta_{2}\;|\xi^{B}\rangle \langle \xi^{B}|+\sin^{2}\theta_{2}\;|\xi^{A}\rangle \langle \xi^{A}|\;. \end{array}$$Then, von Neumann entropy is the Shannon entropy of this mixed state. While this is of interest to much research, Shannon entropy of pure states precedes von Neumann entropy. Also, von Neumann entropy obtained by tracing out pure states has some similarity to the entropy of the superposition of two states Equation (Equation 35), but neglecting both quantities the interference and the conjugate variable of the phase space. Geiger and Kedem [23] showed invariant properties of the Shannon entropy in phase space that von Neumann entropy would not have.

## 6. Conclusions

The Bayesian statistic view was developed to construct the phase space probability density. The Bayesian approach to describe quantum physics in phase space considered the conditional probability densities and joint probability densities to be descriptions of the degree of belief about the phase space variables. It was observed that events representing measurements in the phase space do not occur. However, in a coarse description of the phase space, where the elementary volumes satisfy the uncertainty principle, they do occur. This derivation of the phase space density provides further support to the work of Geiger and Kedem [23,24] where they developed the entropy in phase space.

As a Bayesian theory, quantum physics is subjective, even if the prior knowledge is common to different observers, the measurements may not be. The entanglement scenarios where teleportation experiments have been reported, demonstrate the subjectivity of quantum physics as different observers make different predictions about the outcome of a variable. Moreover, the Bayesian view of quantum physics is that the theory is not a complete causal theory. The time evolution of a state via the unitary evolution is causal, as the Hamiltonian causes the state to evolve. However, the Bayesian combination of observation and the prior knowledge is not causal.

The Bayesian theory of quantum physics also considered the probabilistic object of manipulations to be the state and not the probabilities. Addition and multiplication operations usually associated with **or** and **and** logical operations were applied to quantum states and not to quantum probabilities.

With the phase space probability density constructed, the next objective was to quantify interference and entanglement in terms of information loss or gain, using the Kullback–Liebler divergence (rooted on Shannon entropy). A comparison to the Shannon entropy of the state and some similarities between the two quantities were revealed. One advantage of entropy is that it can be inferred from the quantum state, without any reference to the two states that were used to compute the divergence. It was noted that for spin phase space the relations between entropy and the Kullback–Liebler diverges will differ since the eigenstates of the total spin operator become entangled states.

Extrapolating the Bayesian approach to a philosophical interpretation, and revisiting the Bohr vs. Einstein debate, the view put forward here is similar to the epistemological view of Bohr: quantum theory is today the best model for predictions. The view put forward here also resonates with the ontological concern of Einstein adding the question: “Is quantum theory a causal theory?” If not, ’is there a causal theory to be discovered?” After all, by adopting the Bayesian view, the question of a complete causal theory was left open.

## Figures and Tables

**Figure 1 entropy-25-01227-f001:**
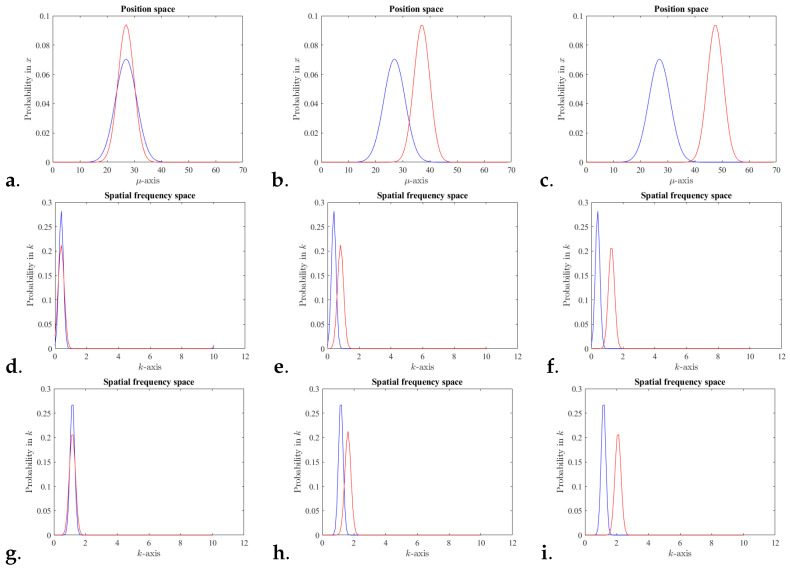
Normal distribution in phase space for two coherent states, **A** and **B** in 1D with centers and variances as follows. For all (**a**–**c**) position space probabilities, with **$\mu_{A}=27$** with **$\sigma_{A}=5.6$** and for (**a**) **$\mu_{B}=27$**, (**b**) **$\mu_{B}=37$**, (**c**) **$\mu_{B}\approx 48$**, all with **$\sigma_{B}=4.2$**. Note that for each coherent state, the spatial frequency value is the phase of the coherent state in position space. For (**d**–**f**), spatial frequency space probabilities, with **$k_{A}\approx 0.35$** and (**d**) **$k_{B}\approx 0.35$**, (**e**) **$k_{B}\approx 0.75$**, (**f**) **$k_{B}\approx 1.15$**. For (**g**–**i**), spatial frequency space probabilities, with **$k_{A}\approx 1.04$** and (**g**) **$k_{B}\approx 1.04$**, (**h**) **$k_{B}\approx 1.55$**, (**i**) **$k_{B}\approx 2.05$**.

**Figure 2 entropy-25-01227-f002:**
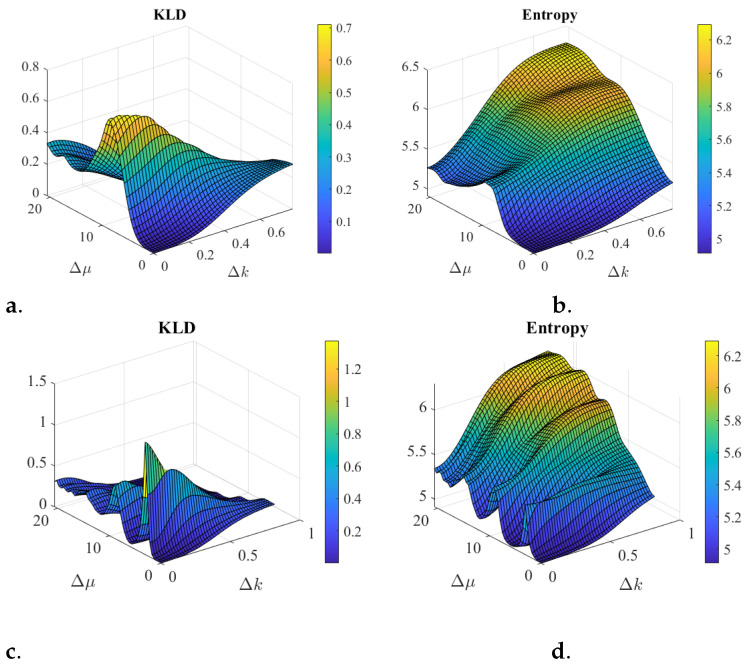
Interference Simulations for a superposition of two coherent states as shown in Figure 1. The coherent state $|\Psi^{A}\rangle$ has $\mu_{A}=27$, $\sigma_{A}=5.6$, and for (**a**,**b**) the phase is $k_{A}=0.35$ while for (**c**,**d**), the phase is $k_{A}=1.04$. The other coherent state $|\Psi^{B}\rangle$ in position has fixed $\sigma_{B}=4.2$ and the position center and phase vary in 48 increments each, as follows: $\mu_{B}=\left[27,\ldots ,48\right]$, and for (**a**,**b**), the phase varies as $k_{B}=\left[0.35,\ldots ,1.15\right]$, while for (**c**,**d**), the phase varies as $k_{B}=\left[1.04,\ldots ,2.05\right]$. The plots axis are all with $\Delta \mu =\mu_{B}-\mu_{A}$ vs. $\Delta k=k_{B}-k_{A}$. The KLD and the entropy become small as the two states closely overlap, i.e., where δμ≈δk≈0. However, the KLD becomes small as the states do not overlap while the entropy gets to be larger. As the phase increases from (**a**,**b**) to (**c**,**d**) oscillation increases for both (KLD and Entropy) as periods reduce. Entropy seems to be a good estimation for the interference behavior when the two states overlap either in spatial frequency or in position. However, the more the overlap in both spaces is reduced the more the two quantities differ in behavior.

**Figure 3 entropy-25-01227-f003:**
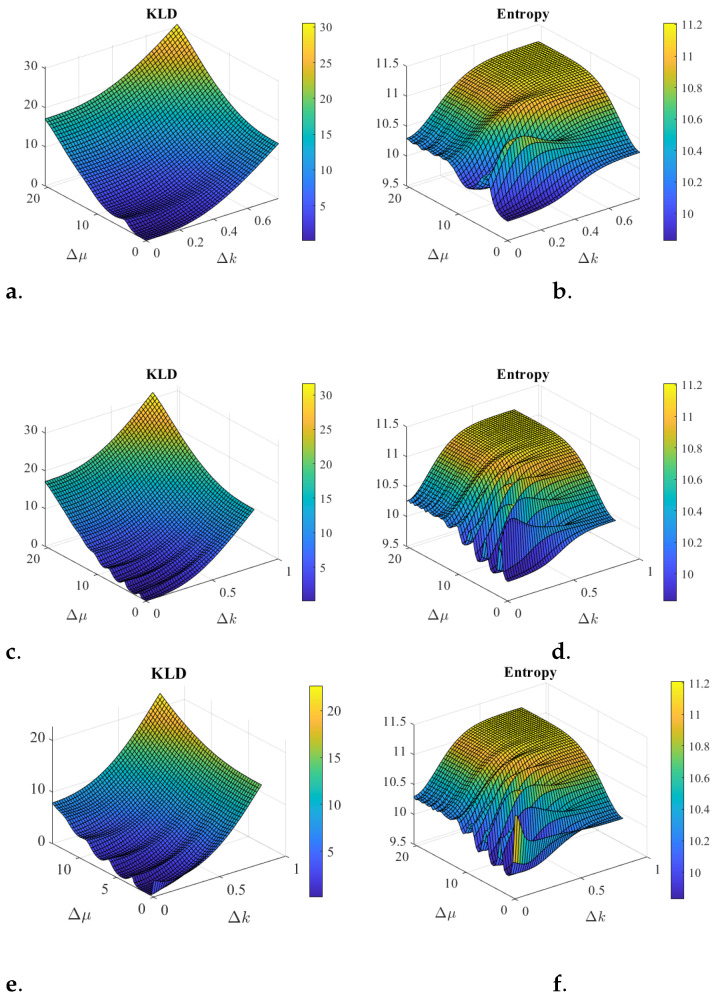
Entanglement simulations from two coherent states shown in Figure 1, with normal probability distributions. Note that the phase of the coherent state projected to position space is the center of the coherent state projected in the spatial frequency space, and vice versa. The coherent state $|\Psi^{A}\rangle$ has a fixed set of parameters, $\mu_{A}=27$, $\sigma_{A}=5.6$ in position space, and for (**a**,**b**) the phase is $k_{A}=0.35$ while for (**c**–**f**) the phase is $k_{A}=1.04$. The coherent state $|\Psi^{B}\rangle$ in position has fixed $\sigma_{B}=4.2$ and the center and phase vary in 48 increments each, as follows: $\mu_{B}\in \left[27,\ldots ,48\right]$, and for (**a**,**b**) $k_{B}\in \left[0.35,\ldots ,1.15\right]$, while for (**c**–**f**) the phase varies as $k_{B}\in \left[1.04,\ldots ,2.05\right]$. The parameter $\theta_{2}=\frac{pi}{4}$ is fixed when entangling the two states. Cases (**a**–**d**) show KLD and Entropy, respectively, for a symmetric entanglement where phase $\varphi_{2}=0$. Cases (**e**,**f**) show KLD and Entropy, respectively, for an anti-symmetric entanglement where phase $\varphi_{2}=\pi$. The effect of the phase $\varphi_{2}$ is only noticeable when the two states are very similar to each other and then both, KLD and entropy, yield large values for the anti-symmetric case (after all anti-symmetric functions must vanish in these cases, while the product of states does not). While the KLD has a smoother behavior, both increase as the separation of the two coherent state parameters increases. The larger values of the phase parameters in (**c**–**f**) clearly cause a periodic behavior. Entropy behavior seems to be a good estimation for entanglement.

## Data Availability

The only data used here is shown in Figure 1 and all the parameters specification to generate the data is described in the caption.

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
