# Peer review of "Quantum Knowledge in Phase Space"

_entropy, 2023, doi:10.3390/e25081227_

Round 1

Reviewer 1 Report

Page 3

For the entire paper the chosen units set h = 1. Then the spatial frequency variable and the quantum momentum variable will have the same value. 

What does it mean, can you pls clarify, do you mean that both basis are the same?

Page 4

Clearly, ρt(x, k) = ρx(xρk(k) is not the product of two independent random variables due to the Fourier transform constraint between the two density functions. 

Is it not what is indicated in the Theorem 1 since,

ρt(x,k) = |ψt(x)||φt(k)| ρx(xρk(k) ?

Or do I miss something?

And what are the consequences of the Theorem 1?

Page 11, Conclusion

One advantage of the entropy is that it can be inferred from the quantum state, without any reference to the two states that were used to compute the divergence. 

Can you pls. Clarify, what do you mean by it? Are there some other consequences?

Is the following statement true or related?

Shannon Entropy specifies “the distribution” of the states, the higher the Entropy there more the states are in a superposition with maximal value present in the uniform distribution.

Author Response

Thank you very much for your review. I have improved the introduction. I believed that it is better organized and the examples I added clarify the concepts put forward. In special I eleaborated more clearly  why Bayesian statistics is a way to viewr quantum physics. Also,  in doing so, I also clarified the section about phase space so that your valid question is better explained.  Also I divided the theorem 1 into a lemma 1 and then the theorem 1. 

I also clarified that the phase space is about spatial frequency and position, and so no need to set hbar=1.  That was confusing, I agree. 

The conclusion was also improved to account for these topics clarified in the introduction about Bayesian statistics  and phase space. 

Of course, the main results and sessions organization remains the same.  

Reviewer 2 Report

The Kullbak Liebler divergence quantifies the distinguishability or distinguishability between two quantum states. For pure states, the KL divergence can capture differences related to interference and entanglement, providing valuable information about the quantum system's properties. Accordingly, Shannon entropy and Kullback-Liebler divergence for pure states in quantum phase spaces allow us to characterize uncertainty, interference, and entanglement in quantum systems. These concepts have applications in quantum information theory, quantum computing, and quantum communication, where understanding and quantifying quantum correlations are essential.
This interesting article provides an illuminating discussion of the above mentioned issues from the Bayesian statistic viewpoint so as to construct a phase space probability density  that is subjective to the observer. In this formulation, quantum physics is a theory of information but not a causal one, whose goal is to quantify interference.

The author provides an enlightening discussion.

I warmly reccommend acceptance

Author Response

Thank you very much for your review. I have improved the introduction. I believed that it is better organized and the examples I added clarify the concepts put forward. In special I eleaborated more clearly  why Bayesian statistics is a way to viewr quantum physics. Also,  in doing so, I also clarified the section about phase space so that your valid question is better explained.  Also I divided the theorem 1 into a lemma 1 and then the theorem 1. 

The conclusion was also improved to account for these topics clarified in the introduction about Bayesian statistics  and phase space. 

Of course, the main results and sessions organization remains the same.  

Round 2

Reviewer 1 Report

Paper is ready for publication.